



# Large contribution of soil N₂O emission to the global warming potential of a large-scale oil palm plantation despite changing from conventional to reduced management practices

Guantao Chen[1], Edzo Veldkamp[1], Muhammad Damris[2], Bambang Irawan[3], Aiyen Tjoa[4], Marife D. Corre[1]

[1]Soil Science of Tropical and Subtropical Ecosystems, Faculty of Forest Sciences and Forest Ecology, University of Goettingen, Göttingen 37077, Germany

[2]Faculty of Science and Technology, University of Jambi, Jl. Raya Jambi-Ma. Bulian km. 15, Mendalo Darat, Muaro, Jambi 36361, Indonesia

[3]Forestry Faculty, University of Jambi, Campus Pinang Masak Mendalo, Jambi 36361, Indonesia

[4]Faculty of Agriculture, Tadulako University, Jl. Soekarno Hatta, km 09 Tondo, Palu 94118, Indonesia

*Correspondence to*: Guantao Chen (gchen1@gwdg.de)

**Abstract.** Conventional management of oil palm plantations, involving high fertilization rate and herbicide application, result in high yield but with large soil greenhouse gas (GHG) emissions. This study aimed to assess a practical alternative to conventional management, namely reduced fertilization with mechanical weeding, to decrease soil GHG emissions without

sacrificing production. We established a full factorial experiment with two fertilization rates (conventional and reduced fertilization, equal to nutrients exported via fruit harvest) and two weeding methods (herbicide and mechanical), each with four replicate plots, since 2016 in a ≥ 15-year old, large-scale oil palm plantation in Indonesia. Soil $CO_2$, $N_2O$, and $CH_4$ fluxes were measured during 2019 – 2020 and yield was measured during 2017 – 2020. Fresh fruit yield (30 ± 1 Mg ha⁻¹ yr⁻¹) and soil GHG fluxes did not differ among treatments ($P \geq 0.11$), implying legacy effects of over a decade of conventional management

prior to the start of experiment. Annual soil GHG fluxes were 5.5 ± 0.2 Mg $CO_2$-C ha⁻¹ yr⁻¹, 3.6 ± 0.7 kg $N_2O$-N ha⁻¹ yr⁻¹, and −1.5 ± 0.1 kg $CH_4$-C ha⁻¹ yr⁻¹ across treatments. The palm circle, where fertilizers are commonly applied, covered 18% of the plantation area but accounted 79% of soil $N_2O$ emission. The net primary production of this oil palm plantation was 17150 ± 260 kg C ha⁻¹ yr⁻¹ but 62% of this was removed by fruit harvest. The global warming potential of this planation was 3010 ± 750 kg $CO_2$-eq ha⁻¹ yr⁻¹ of which 55% was contributed by soil $N_2O$ emission and only < 2% offset by soil $CH_4$ sink.

## 25 1 Introduction

With increasing demand for vegetable oil, oil palm as a productive woody oil crop is widely planted in the tropics (Descals et al., 2021). Globally, the oil palm-planted area rapidly increased from 4 million ha in 1980 to 28 million ha in 2019 (FAO, 2021), and oil palm plantations are expected to continue to expand to meet the increasing demand of a growing world population (OECD, 2022). Oil palm expansion drives tropical deforestation (Vijay et al., 2016) and is accompanied by serious reductions

in multiple ecosystem functions, e.g. decreases in C storage (Kotowska et al., 2015; van Straaten et al., 2015), nutrient cycling and retention (Allen et al., 2015; Kurniawan et al., 2018), and biodiversity losses (Clough et al., 2016). Despite these losses of ecosystem multifunctionality, profit gains increase under oil palm plantations (Grass et al., 2020), which increase farm and non-farm households' income and improve their livelihood (Bou Dib et al., 2018). Moreover, the high yield of oil palm plays an invaluable role in meeting human demand for vegetable oil (Thomas et al., 2015; Rochmyaningsih, 2019). However, there



is a need for a balance between economic gains and maintaining or avoiding further degradation of ecosystem functions (Bessou et al., 2017). Conventional management practices of large-scale oil palm plantations (> 50 ha planted area, can be up to 20000 ha and owned by corporations), particularly high fertilization rates and herbicide application, are agents of these decreases in ecosystem functions (Tao et al., 2016; Ashton-Butt et al., 2018; Rahman et al., 2019). Thus, there is a need for practical solutions that can easily be implemented in order to reduce these negative impacts on ecosystem functions without sacrificing
productivity and profit.

Soil greenhouse gas (GHG) emissions are a concern in oil palm plantations (Kaupper et al., 2020; Skiba et al., 2020). Compared to forests, the reductions in plant biomass production, soil organic carbon (SOC) and soil microbial biomass as well as C removal from the field via fruit harvest and increase in soil bulk density in oil palm plantations largely decrease the latter's GHG abatement capability (Kotowska et al., 2015; van Straaten et al., 2015; Clough et al., 2016). This capability is influenced
by agricultural management practices, especially fertilization rates (Sakata et al., 2015; Hassler et al., 2017; Rahman et al., 2019). N fertilization in oil palm plantations during the wet season can increase soil $N_2O$ emissions (Aini et al., 2015; Hassler et al., 2017), a potent GHG and agent of ozone depletion (Davidson et al., 2000). Soil $N_2O$ emissions from tropical agriculture is largely controlled by soil mineral N availability, which in turn is influenced by N fertilization rate, and soil moisture, as $N_2O$-production processes of denitrification and nitrification in the soil are favored under high soil mineral N and moisture levels
(Khalil et al., 2002; Liyanage et al., 2020; Quiñones et al., 2022). In Indonesia, a large-scale oil palm plantation with commonly high N fertilization rate has higher soil $N_2O$ emissions than smallholder oil palm plantations (< 5 ha of planted area per household) with low N fertilization rates (Hassler et al., 2017).

Soil $CO_2$ emissions originate from heterotrophic and root respiration (Bond-Lamberty et al., 2004). The temporal pattern of soil $CO_2$ emission follows the seasonal dynamics of soil moisture and/or soil temperature, exhibiting low soil $CO_2$ emissions
at low soil moisture content, increases toward an optimum soil moisture, and decrease towards water saturation when oxygen availability and gas diffusion limit soil $CO_2$ emissions (Sotta et al., 2007; van Straaten et al., 2011). The spatial pattern of soil $CO_2$ emissions in tropical ecosystems is influenced by the spatial variation in soil organic matter, bulk density, root biomass and available N and P levels in the soil (Adachi et al., 2006; Hassler et al., 2015; Cusack et al., 2019; Tchiofo Lontsi et al., 2020). Another important GHG that is influenced by management practices in tropical plantations or croplands is $CH_4$ (Hassler
et al., 2015; Quiñones et al., 2022). Soil surface $CH_4$ flux is the net effect of $CH_4$ production by methanogenic archaea and $CH_4$ oxidation by methanotrophic bacteria (Hanson and Hanson, 1996). In well-drained tropical soils, $CH_4$ oxidation usually is more dominant than $CH_4$ production, resulting in a net soil $CH_4$ uptake or negative $CH_4$ flux (Veldkamp et al., 2013). Seasonal pattern of soil surface $CH_4$ flux in smallholder oil palm plantations in Indonesia reflects the seasonal variation of soil moisture, with lower $CH_4$ uptake during the wet season than the dry season (Hassler et al., 2015). In well-drained tropical soils with low
N availability, soil $CH_4$ uptake or methanotrophic activity is enhanced with increase in soil mineral N content (Veldkamp et al., 2013; Hassler et al., 2015; Tchiofo Lontsi et al., 2020). However, in tropical agricultural soils with high soil mineral N levels ($NH_4^+$ and $NO_3^-$) from high N fertilization, competition of $NH_4^+$ against $CH_4$ for the active site of mono-oxygenase enzyme can reduce soil $CH_4$ uptake (Hanson and Hanson, 1996). Veldkamp et al. (2001) observed a temporary inhibition of $CH_4$ uptake for approximately three weeks following $NH_4^+$-based fertilizer application in tropical pasture soils. In contrast,
high soil $NO_3^-$ level (e.g. resulting from nitrification of applied N fertilizer) can inhibit $CH_4$ production in the soil since $NO_3^-$ is preferred over bicarbonate as an electron acceptor (Martinson et al., 2021; Quiñones et al., 2022). Nonetheless, across forests,



smallholder rubber and oil palm plantations in Indonesia, the overriding pattern is the increase in soil $CH_4$ uptake with increase in soil mineral N, suggesting the prevailing control of N availability on methanotrophic activity in the soil (Hassler et al., 2015). The spatial patterns of soil surface $CH_4$ fluxes depict the spatial variations in soil properties that affects soil moisture

content and gas diffusivity, such as soil texture (Veldkamp et al., 2013; Tchiofo Lontsi et al., 2020), soil bulk density and organic matter (e.g. at a plot or landscape scale; Tchiofo Lontsi et al., 2020).

In large-scale oil palm plantations, the typical management practices of fertilization, weeding, and pruning of senesced fronds result in three distinctive spatial management zones: palm circle (weeded and fertilizers are applied), inter-row (weeded but not fertilized) and frond-stacked area (where pruned fronds are piled) (Fig. 1; Formaglio et al., 2021). In the palm circle and

inter-row, frequent management activities (weeding, pruning and harvesting) result in soil compaction by foot traffic (increased soil bulk density) and the low litter input in these zones exhibits low SOC and microbial biomass, and low soil N cycling rate (Formaglio et al., 2021). Additionally, root biomass is high in the palm circle (Dassou et al., 2021). In the frond-stacked area, decomposition of fronds results in large SOC (with decreased soil bulk density), large microbial and fine root biomass, and high soil N cycling rate (Moradi et al., 2014; Rüegg et al., 2019; Formaglio et al., 2020; Dassou et al., 2021). Overall, the

differences in soil properties and root biomass among these spatially distinct management zones (Formaglio et al., 2021) potentially drive the spatial variation of soil GHG fluxes from oil palm plantations (Hassler et al., 2015, 2017; Aini et al., 2020). Thus, estimating soil GHG emissions from oil palm plantations should take into account the spatial variability among management zones within a site or plot.

This study aimed to (1) assess differences in soil GHG fluxes from conventional high fertilization rates with herbicide

application compared to alternative management of reduced fertilization rates (equal to nutrient exported via fruit harvest) with mechanical weeding; and (2) determine the controlling factors of the soil GHG fluxes from a large-scale oil palm plantation. A $2 \times 2$ factorial field experiment with conventional and reduced fertilization rates as well as herbicide and mechanical weed control was established in a $\geq 15$-year old, large-scale oil palm plantation in Jambi, Indonesia starting in November 2016. The earlier studies during the first 1.5 years of this oil palm management experiment show comparable gross rates of soil N cycling,

microbial biomass (Formaglio et al., 2021), and root biomass (Ryadin et al., 2022) among treatments. Thus, we hypothesized that during 2.5–3.5 years of this management experiment, the reduced fertilization with mechanical weeding will have comparable soil $CO_2$ and $CH_4$ fluxes but lower soil $N_2O$ emissions than the conventional fertilization with herbicide weeding. Moreover, we hypothesized that the fertilized palm circle that has high soil bulk density and root biomass but low SOC, soil microbial biomass and N cycling rate (Dassou et al., 2021; Formaglio et al., 2021) will have large soil $CO_2$ and $N_2O$ emissions

but small soil $CH_4$ uptake. The unfertilized inter-row that has high soil bulk density but low SOC, microbial biomass and N cycling rate (Formaglio et al., 2021) will have small soil $CO_2$, $N_2O$ emissions and $CH_4$ uptake. The frond-stacked area (i.e. unfertilized but piled with pruned fronds) that has large SOC, microbial biomass and N cycling rate but low soil bulk density (Formaglio et al., 2021) will have large soil $CO_2$ emissions and $CH_4$ uptake but small soil $N_2O$ emissions. In this study, we assessed the soil GHG footprint and the global warming potential (GWP) of a typical large-scale oil palm plantation in order

to evaluate reduced management (i.e. reduced fertilization rate with mechanical weeding) against the commonly employed conventional management (i.e. high fertilization rate with herbicide application).



## 2 Materials and methods

### 2.1 Site description and experimental design

This study was conducted in a large-scale oil palm plantation in Jambi, Indonesia (1°43′8″ S, 103°23′53″ E, 73 m above sea level). The plantation was 2025 ha and established between 1998 and 2002, and thus was ≥ 18 years old during our measurement of soil GHG fluxes from July 2019 to June 2020. The oil palms were planted in a triangular pattern with 8 m spacing between palms and the planting density was 142 palms ha$^{-1}$. Mean annual (2010–2020) air temperature is $26.9 \pm 0.2$ °C and mean annual precipitation is $2078 \pm 155$ mm. The soil is Acrisol with a sandy clay loam texture (Table S1). More than 18 years of management induced three distinct zones within this oil palm plantation: palm circle, inter-row, and frond-stacked area (Fig. 1) (Formaglio et al., 2020). The palm circle (a 2 m radius from the palm base) is the zone where fertilizers and lime are applied (in April and October of each year) and is weeded every three months; this represents 18% of the plantation area. The inter-row is unfertilized but weeded every six months; this represents 67% of the plantation area. The frond-stacked area is where senesced fronds are piled and is neither fertilized nor weeded; this represents 15% of the plantation area. As consequences of these management activities, SOC and total N stocks are higher whereas soil bulk density is lower in the frond-stacked area than the palm circle and inter-row. The effective cation exchange capacity and pH, influenced by the applied lime as well as from decomposed leaf litter, are higher in the palm circle and frond-stacked area than the inter-row (Table S1) (Formaglio et al., 2020).

This oil palm management experiment had started in November 2016 – a 2 × 2 factorial design of two fertilization rates and two weeding methods: conventional fertilization – herbicide weeding (ch), conventional fertilization – mechanical weeding (cw), reduced fertilization – herbicide weeding (rh), and reduced fertilization – mechanical weeding (rw). These four treatments were randomly assigned to four plots (50 m × 50 m each) in a block and there were four replicate blocks (Fig. 1). Within each plot, we selected two subplots in the inner 30 m × 30 m area and each subplot included the three management zones where all measurements were carried out (Fig. 1).

Conventional fertilization rates were 260 kg N, 50 kg P, and 220 kg K ha$^{-1}$ yr$^{-1}$, commonly practiced in large-scale oil palm plantations in Jambi, Indonesia (Formaglio et al., 2020). Reduced fertilization rates were 136 kg N, 17 kg P, and 187 kg K ha$^{-1}$ yr$^{-1}$, equal to the nutrients exported by fruit harvest (detail calculation given by Formaglio et al., 2021). The fertilizer sources were urea, triple superphosphate, muriate of potash or NPK-complete. For herbicide weed control, glyphosate was used at a rate of 1.5 L ha$^{-1}$ yr$^{-1}$ in the palm circle (split into four applications yr$^{-1}$) and 0.75 L ha$^{-1}$ yr$^{-1}$ in the inter-row (split into two applications yr$^{-1}$). Mechanical weeding used a brush cutter with the same weeding frequencies as the herbicide applications. All treatments received the same rates of lime (426 kg dolomite ha$^{-1}$ yr$^{-1}$) and micronutrients (142 kg micro-mag ha$^{-1}$ yr$^{-1}$ with 0.5% $B_2O_3$, 0.5% CuO, 0.25% $Fe_2O_3$, 0.15% ZnO, 0.1% MnO and 18% MgO), applied only in the palm circle.

### 2.2 Soil greenhouse gas fluxes

Soil $CO_2$, $N_2O$, and $CH_4$ fluxes were measured monthly from July 2019 to June 2020, using vented static chambers. Measurement schedules were random among plots (i.e. 16 plots, each with two subplots that each encompassed three management zones) such that we covered the temporal variability of soil GHG fluxes (e.g. peak of soil $N_2O$ emissions during two weeks following fertilization), as observed in our earlier study (Hassler et al., 2017). During the year-round measurements,





chamber bases (0.04 m$^2$ area) were installed permanently at each management zone (i.e. palm circle, inter-row, and frond-stacked area) in all subplots of 16 plots (Fig. 1), totaling to 96 chambers, by inserting these into the soil at approximately 0.02 m depth. On a measurement day, the chamber bases were covered for 28 minutes with polyethylene covers (11 L total volume)

that were equipped with a Luer-lock sampling port. Four gas samples (23 mL each) were taken using syringes at 1, 10, 19, 28 minutes following chamber closure and injected into pre-evacuated 12 mL glass vials (Labco Exetainers, Labco Limited, Lampeter, UK) with rubber septa. On each monthly measurement, 384 gas samples were taken (i.e. 16 plots × 2 subplots × 3 management zones × 4 chamber headspace-sampling intervals). As a check for possible leakage, we also stored standard gases into pre-evacuated 12 mL glass vials in the same period as the field gas samples. All gas samples were transported to the

Goettingen University, Germany for analysis.

Gas samples were analyzed using a gas chromatograph (SRI 8610C, SRI Instruments Europe GmbH, Bad Honnef, Germany) equipped with a flame ionization detector to measure $CH_4$ and $CO_2$ concentrations (with a methanizer) as well as an electron capture detector for $N_2O$ analysis (with a make-up gas of 5% $CO_2$–95% $N_2$). Soil $CO_2$, $N_2O$, and $CH_4$ fluxes were calculated from the linear change in concentrations with chamber closure time, adjusted with the measured air temperature and

atmospheric pressure during sampling. We found that the concentrations of all standard gases stored in the same duration as the field gas samples stayed the same as those in the standard gases at our laboratory. The quality check for each flux measurement was based on the linear increase of $CO_2$ concentrations with chamber closure time ($R^2 \geq 0.9$). For soil $CH_4$ and $N_2O$, all flux measurements (including zero and negative fluxes) were included in the data analysis. For an overall value of soil $CO_2$, $N_2O$, and $CH_4$ fluxes in a plot, fluxes were weighted by the areal coverages of the three management zones (see above).

Area-weighted annual soil $CO_2$, $N_2O$, and $CH_4$ fluxes were estimated based on trapezoidal extrapolations between measured fluxes and sampling day intervals.

### 2.3 Soil variables

Concurrent with soil GHG flux measurement, soil temperature, mineral N, and moisture content in top 5 cm depth were determined. Soil temperature was recorded using a portable thermometer (Greisinger GMH 3210, Greisinger Messtechnik

GmbH, Regenstauf, Germany). At about 1 m away from each chamber, soil samples were collected and pooled from the two subplots for each management zone. Part of each soil sample was added to a prepared bottle containing 150 mL 0.5 M $K_2SO_4$ for immediate mineral N extraction. Upon arrival at the field laboratory, the bottles were shaken for 1 hour, filtered and the extracts were immediately frozen. The remaining soil sample was oven-dried at 105 °C for 24 h to determine the gravimetric moisture content, which was used to calculate the dry mass of the soil extracted for mineral N. The moisture content was the

expressed as water-filled pore space (WFPS), using the mineral soil particle density of 2.65 g cm$^{-3}$ and the average soil bulk density in the top 5 cm (1.23 g cm$^{-3}$ in the palm circle, 1.20 g cm$^{-3}$ in the inter-row, and 0.52 g cm$^{-3}$ in the frond-stacked area). In April and May 2020, we were unable to conduct WFPS and mineral N measurements due to restrictions from COVID-19 pandemic. The frozen extracts were transported by air to Goettingen University and analyzed for $NH_4^+$ and $NO_3^-$ concentrations using continuous flow injection colorimetry (SEAL Analytical AA3, SEAL Analytical GmbH, Norderstedt, Germany).

### 2.4 Global warming potential estimation

The GWP of this ≥ 18-year old, large-scale oil palm plantation was estimated based on Malhi et al. (1999), as also used in our





earlier work in agricultural land use (Quiñones et al., 2022). First, the net primary production (NPP) was estimated as the sum of aboveground biomass C production, fruit biomass C production, frond litter biomass C input, root biomass C production, and root litter biomass C input. Within the inner 30 m × 30 m area per plot, stem height, harvested fruit weight, and the number of pruned fronds per palm were recorded during 2017–2020 (Iddris et al., 2023). Aboveground biomass per palm was calculated using the allometric growth equation (Asari et al., 2013): kg biomass palm$^{-1}$ = 71.797 × palm stem height – 7.0872; and biomass production per palm was the difference in biomass between two consecutive years. Annual aboveground biomass C production (g C m$^{-2}$ yr$^{-1}$) = annual biomass production per palm (kg palm$^{-1}$, 2019–2020) × planting density (142 palms ha$^{-1}$) × tissue C concentration (0.41 g C g$^{-1}$) × 10$^{-1}$ (for unit conversion). Annual fruit biomass C production (g C m$^{-2}$ yr$^{-1}$) = annual fruit harvest per palm (kg palm$^{-1}$, mean of 2019–2020) × planting density × tissue C concentration (0.63 g C g$^{-1}$) × 10$^{-1}$. Frond litter biomass C input (g C m$^{-2}$ yr$^{-1}$) = annual litter production per palm (kg palm$^{-1}$, mean of 2019–2020) × planting density × tissue C concentration (0.47 g C g$^{-1}$) × 10$^{-1}$. Data of root biomass C production (140 g C m$^{-2}$ yr$^{-1}$) and root litter biomass C input (45 g C m$^{-2}$ yr$^{-1}$) were taken from Kotowska et al. (2015).

Second, the net ecosystem productivity (NEP) was calculated as NEP (g C m$^{-2}$ yr$^{-1}$) = heterotrophic respiration – (NPP – fruit biomass C) (Malhi et al., 1999; Quiñones et al., 2022). Our measured soil $CO_2$ fluxes included both autotrophic and heterotrophic respirations. We assumed 70% heterotrophic contribution to soil $CO_2$ flux, based on a long-term quantification in a forest in Sulawesi, Indonesia (van Straaten et al., 2011). As the frond litter also contributes to heterotrophic respiration upon decomposition, we assumed this fraction to be 80% of frond litter biomass C, based on the frond-litter decomposition rate in the same plantation (Iddris et al., 2023). Third, the GWP (g $CO_2$-eq m$^{-2}$ yr$^{-1}$) = (NEP × 3.67) + (soil $N_2O$ fluxes × 298) + (soil $CH_4$ fluxes × 25), of which 3.67 is C-to-$CO_2$ conversion, and 298 and 25 are $CO_2$-equivalents of $N_2O$ and $CH_4$, respectively, for a 100-year time horizon (IPCC, 2006). Negative and positive symbols indicate the direction of the flux: (−) for C uptake and (+) for C export or emission from the plantation.

**2.5 Statistical analysis**

The mean value of two subplots for the soil GHG fluxes and soil temperature were used to represent each plot and management zone on each sampling day. The normality of distribution and equality of variance were first tested using Shapiro-Wilk's test and Levene's test, respectively. Linear mixed-effects (LME) models with Tukey's HSD test were used to assess the differences in soil GHG fluxes and soil variables (WFPS, soil temperature, and mineral N content) among treatments (Crawley, 2013). In the LME models, management (2 × 2 factorial of fertilization rates, weed control and their interaction) was considered as the fixed effect whereas plot and sampling day were taken as random effects, and statistical analysis were conducted for each management zone. As there were no significant differences among treatments (Table 1), we tested differences among the three management zone across treatments; for the latter, management zone was the fixed effect in the LME model and plot and sampling day were random effects. We also assessed if there were seasonal differences, and the year-round measurements were categorized into dry (precipitation ≤ 80 mm month$^{-1}$) and wet seasons; this was conducted for each management zone, and season was the fixed effect in the LME model and plot and sampling day were random effects. For all the above analyses, the LME models further included a variance function that allows variance heteroscedasticity of the fixed effect, and/or a first-order temporal autoregressive process that assumes decreasing auto-correlation between sampling days with increasing time difference, if these improve the model performance based on Akaike information criterion. The model residual was checked



using diagnostic plots and finally soil $CO_2$ and $N_2O$ fluxes, and soil $NH_4^+$ and $NO_3^-$ concentrations were re-analyzed after log-transformation as the model residual distributions approximated the normal distribution.

The relationships between soil GHG fluxes and soil variables were determined by Spearman's Rank correlation test. Correlations tests were conducted on the means of the four replicate plots on each measurement day for each management zone ($n = 144$ for soil temperature, from 4 treatments × 3 management zones × 12 monthly measurements; $n = 120$ for WFPS and mineral N content, from 4 treatments × 3 management zones × 10 monthly measurements). All data analyses were performed using the R version 4.0.5 (R core Team, 2021). The statistical significance for all the tests was set at $P \leq 0.05$.

## 3 Results

### 3.1 Soil greenhouse gas fluxes and global warming potential estimate

Soil $CO_2$ emissions from the palm circle and frond-stacked area were higher in the wet season than in the dry season ($P \leq 0.03$; Fig. S1). Soil $N_2O$ emissions from the palm circle sharply increased after fertilizer application and returned to background levels after two months (Fig. S2). Excluding the direct effects after fertilization, the palm circle had higher soil $N_2O$ emissions

in the wet season than the dry season ($P = 0.03$; Fig. S2). Soil $CH_4$ uptake was higher in the dry season than the wet season in all three management zones ($P \leq 0.01$; Fig. S3). The frond-stacked area showed consistent $CH_4$ uptake throughout the measurement period whereas 33% and 17% of measured soil $CH_4$ fluxes in the palm circle and inter-row, respectively, were net $CH_4$ emissions (Fig. S3).

Reduced and conventional management had comparable soil $CO_2$ emissions (fertilization, weeding and interaction: $P \geq 0.13$),

$N_2O$ emissions (fertilization, weeding and their interaction: $P \geq 0.14$), and $CH_4$ fluxes (fertilization, weeding and their interaction: $P \geq 0.26$) in each management zone (Table 1). However, there were clear differences in soil GHG fluxes among the three management zones. The palm circle had the highest soil $N_2O$ emissions and lowest soil $CH_4$ uptake ($P \leq 0.01$; Table 1); the inter-row had the lowest soil $CO_2$ and $N_2O$ emissions ($P \leq 0.01$; Table 1); the frond-stacked area had the highest soil $CO_2$ emissions and $CH_4$ uptake ($P \leq 0.01$; Table 1). The palm circle accounted for 25%, the inter-row for 45%, and the frond-

stacked area for 30% of the annual soil $CO_2$ emissions (Fig. 2). The palm circle comprised 79% of the annual soil $N_2O$ emissions although it only accounted for 18% of the plantation area (Fig. 2). The frond-stacked area with 15% areal coverage contributed to 41% of the annual soil $CH_4$ uptake and the palm circle with 18% of plantation area contributed 5% of the annual soil $CH_4$ uptake (Fig. 2).

We calculated the GWP across 16 plots (Fig. 3) as there were no significant differences in soil GHG fluxes among treatments

(Table 1). Additionally, the reduced and conventional management had also comparable fruit yield during four years (2017–2020) of treatments (fertilization, weeding and their interaction: $P \geq 0.07$; Table S2). The NPP was larger than the soil heterotrophic respiration (which was assumed to be 70% of the measured soil respiration + 80% C emissions from decomposition of frond litter; see Methods), but 62% of this NPP was removed from the field via fruit harvest (Fig. 3). Thus, this oil palm plantation turned into a net C source (i.e. positive NEP value; Fig. 3). Summing the NEP, soil $N_2O$ emissions, and

soil $CH_4$ uptake in terms of $CO_2$ equivalent (100-year time horizon; see Methods), the GWP of this ≥ 18-year old, large-scale oil palm plantation was contributed by 55% soil $N_2O$ emissions and only counterbalanced by < 2% soil $CH_4$ sink (Fig. 3).



### 3.2 Soil variables

Fertilization and weeding treatments did not affect soil temperature (fertilization, weeding and their interaction: $P \geq 0.21$) and WFPS (fertilization, weeding and interaction: $P \geq 0.24$) in each management zone (Table 2). Soil $NO_3^-$ concentration was

lower in reduced than conventional fertilization, particularly in the frond-stacked area ($P \leq 0.01$; Table 2). There was an interaction effect of fertilization and weeding on soil $NH_4^+$ concentration in the frond-stacked area ($P = 0.02$; Table 2); however, neither fertilization nor weeding solely affect soil $NH_4^+$ concentration in any of the management zones ($P \geq 0.08$; Table 2). Across treatment plots, the three management zones showed comparable soil temperature while the palm circle and inter-row had higher WFPS than the frond-stacked area ($P \leq 0.01$; Table 2). The soil $NH_4^+$ and $NO_3^-$ concentrations in the palm circle

and frond-stacked area were larger than the inter-row ($P \leq 0.01$; Table 2).

Soil $CO_2$ emissions were positively correlated with WFPS in the palm circle (rho = 0.37, $P = 0.02$) and frond-stacked area (rho = 0.72, $P \leq 0.01$; Fig. 4a) and were positively correlated with soil temperature in the frond-stacked area (rho = 0.60, $P \leq 0.01$) and inter-row (rho = 0.29, $P = 0.05$; Fig. 4b). Soil $N_2O$ emissions were positively correlated with soil mineral N in the palm circle (rho = 0.58, $P \leq 0.01$) and inter-row (rho = 0.32, $P = 0.04$; Fig. 4f). Soil $CH_4$ uptake decreased with increase in WFPS in

all three management zones (rho = 0.44–0.81, $P \leq 0.01$; Fig. 4g). In the frond-stacked area, soil $CH_4$ uptake decreased with increase in soil temperature (rho = 0.54, $P \leq 0.01$; Fig. 4h) but increased with increase in soil mineral N (rho = −0.40, $P = 0.01$; Fig. 4i); also in the frond-stacked area, a positive relationship between soil temperature and WFPS was observed (rho = 0.37, $P = 0.02$; Fig. 4j). We did not found any other significant correlations between soil GHG fluxes and the measured soil variables.

## 4 Discussion

### 4.1 Soil $CO_2$ emissions

Area-weighted soil $CO_2$ emissions (Table 1) were only about one-third of the soil $CO_2$ emissions from forests in the same study area (187–196 mg C m$^{-2}$ h$^{-1}$; Hassler et al., 2015) but within the range reported for oil palm plantations on mineral soils in Southeast Asia (45–195 mg C m$^{-2}$ h$^{-1}$; Hassler et al., 2015; Sakata et al., 2015; Aini et al., 2020; Drewer et al., 2021b). Specifically, soil $CO_2$ emissions from the inter-row were in the lower end of this range and soil $CO_2$ fluxes from the frond-

stacked area were in the middle of this range. These earlier studies deployed different spatial sampling designs for measuring soil GHG fluxes from oil palm plantations. Sakata et al. (2015) measured soil $CO_2$ fluxes at 1 m away from the palm base and Hassler et al. (2015) measured soil $CO_2$ fluxes at 1.8–5 m away from the palm base. Aini et al. (2020) measured soil $CO_2$ fluxes from the fertilized area (within 1 m from the palm base) and unfertilized area whereas no information on sampling location was given by Drewer et al. (2021b). Measurement locations to represent spatial management zones should be stated when

reporting soil GHG fluxes from oil palm plantations in order to facilitate comparisons as well as to warrant spatial extrapolation.

The three management zones differed in soil $CO_2$ fluxes caused by their differences in SOC (Table S1), microbial biomass (Fig. S4) as drivers of heterotrophic respiration, and root biomass (Nelson et al., 2014) that influences autotrophic respiration. In this mature large-scale oil palm plantation, senesced fronds have been piled on the frond-stacked area for more than a decade. This results in 40% larger SOC stocks (Table S1) and 3–5 times larger microbial biomass (Fig. S4) in the frond-stacked area than

in the palm circle and inter-row (Formaglio et al., 2020, 2021). The positive correlation of microbial biomass C (MBC) to soil



$CO_2$ fluxes (Fig. S4) supported our second hypothesis whereby differences in soil $CO_2$ emissions among management zones are driven in part by microbial biomass size and available organic C for heterotrophic respiration. Substantial heterotrophic respiration as well as presence of roots in the frond-stacked area (Rüegg et al., 2019; Dassou et al., 2021) explained its highest soil $CO_2$ emissions (Table 1). On the other hand, the palm circle has higher root biomass than the inter-row (Nelson et al., 2014), and higher soil $CO_2$ emissions from the palm circle than the inter-row (Table 1) may be caused by their disparate autotrophic respiration as their soil SOC stocks (Table S1) and MBC did not differ (Formaglio et al., 2021). However, in the same oil palm plantation, the different fertilization and weeding treatments, analyzed across the three management zones, did not influence soil MBC (Formaglio et al., 2021) as well as the fine root biomass in the top 10 cm depth (Ryadin et al., 2022) after one year of this management experiment. These findings support our first hypothesis whereby there was no short-term differences between reduced and conventional managements on soil $CO_2$ emissions. Nonetheless, we emphasize that the lower soil respiration in oil palm plantations compared to the forests (Hassler et al., 2015) is supported by its decreases in SOC (van Straaten et al., 2015; Allen et al., 2016), root and litter production (Kotowska et al., 2015) and microbial biomass (Allen et al., 2015; Formaglio et al., 2021). Also, the higher soil [15]N natural abundance and lower soil C:N ratio in oil palm plantations than the forests (Hassler et al., 2015; van Straaten et al., 2015) signify a highly decomposed organic matter which, combined with reduced SOC, suggest reduced available C for microbial biomass and heterotrophic activity (Allen et al., 2015; Formaglio et al., 2021).

The seasonal pattern of soil $CO_2$ emissions from land uses in Indonesia is commonly influenced by soil moisture (Hassler et al., 2015; van Straaten et al., 2011). In our present study, the positive correlation between soil $CO_2$ emissions and soil moisture (ranging from 10%–55% WPFS; Fig. 4a) depicted a reduced soil respiration during the dry season, particularly in the palm circle and frond-stacked area, suggesting diminished autotrophic and heterotrophic respiration in these management zones when soil moisture was low (Fig. 4a; Fig. S1). Previous studies show that autotrophic and heterotrophic respiration increase toward an optimum WFPS (e.g. WFPS between 50%–55%; Sotta et al., 2007; van Straaten et al., 2011). Unlike the previous study conducted in 2013 in the same area (Hassler et al., 2015), our measured WFPS did not reach beyond 55%, as the annual rainfall during our study year (2019–2020) was lower than in 2013 and the sandy clay loam texture of our present soil may facilitate well-drained conditions. Thus, we did not observe a parabolic relationship of soil $CO_2$ emissions with WFPS beyond 55% as observed by Hassler et al. (2015). Although we observed a positive relationship between soil $CO_2$ emissions and soil temperature (Fig. 4b), largely in the frond-stacked area, this maybe confounded by WFPS as the soil temperature and WFPS were auto-correlated (Fig. 4j). Thus, soil temperature was not a dominant controlling factor for the seasonal pattern of soil $CO_2$ emissions in this oil palm plantation where soil temperature also only varied narrowly during our measurement period (25–28 °C; Fig. 4b).

### 4.2 Soil $N_2O$ emissions

Area-weighted soil $N_2O$ emissions (Table 1) were within the range reported for oil palm plantations on mineral soils (8–117 $\mu g\,N\,m^{-2}\,h^{-1}$; Aini et al., 2015; Sakata et al., 2015; Hassler et al., 2017 ; Rahman et al., 2019; Drewer et al., 2021b). Specifically, the soil $N_2O$ emissions from the unfertilized inter-row and frond-stacked areas at our site were close to the lower end of this range whereas those from the fertilized palm circle were larger than the upper end of this range (Fig. S2). This pattern supported our second hypothesis, whereby soil $N_2O$ emission was primarily influenced by soil N availability (i.e. mineral N; Table 2; Fig.



4f). These pulses of $N_2O$ emissions from the palm circle peaked at around two weeks following N fertilization and went down to the background emissions after at most eight weeks (Fig. S2) (Aini et al., 2015; Hassler et al., 2017; Rahman et al., 2019). Although both the inter-row and frond-stacked areas had no direct N fertilizer application, litter decomposition in the frond-stacked area resulted in higher gross rates of N mineralization and nitrification, indicating higher soil N availability, than the inter-row (Formaglio et al., 2021). This explained the higher soil $N_2O$ emissions from the frond-stacked area than the inter-row (Table 1). However, these internal soil-N cycling processes provide slow release of mineral N as opposed to the pulse release of mineral N level from N fertilization, and hence the soil $N_2O$ emissions and mineral N levels were larger in the palm circle than the frond-stacked area (Tables 1 and 2). These findings signified the main control of soil N availability on soil $N_2O$ emissions (Fig. 4f) (Davidson et al., 2000). We did not observe a correlation of soil $N_2O$ emissions with WFPS (Fig. 4d) possibly because our sandy clay loam soil was relatively dry to moist in the top 5 cm depth ($\leq$ 55% WFPS) during our measurement period. In sum, the direct N fertilizer application on the palm circle (although covering only 18% of the plantation area) caused the extremely high soil $N_2O$ emissions (Fig. S2), accounting 79% of the annual soil $N_2O$ emission at the plantation level (Fig. 2). Our findings highlight that the palm circle was a hot spot of soil $N_2O$ emissions, and such management-induced spatial heterogeneity must be accounted for in accurately quantifying soil $N_2O$ emissions from large-scale oil palm plantations.

The large and comparable soil $N_2O$ emissions between the conventional and reduced fertilization treatments (Table 1; Fig. 2) were contrary to our first hypothesis. However, this finding was consistent with the soil N availability, i.e. mineral N (Table 2) as well as gross and net rates of soil N mineralization and nitrification, which did not differ among treatments during 1–4 years of this management experiment (Formaglio et al., 2021; unpublished data of net N mineralization and nitrification rates). These results implied a substantial legacy effect of the past decade of conventional management (high fertilization rate and herbicide application) prior to the start of this management experiment. It is important to note that our reduced fertilization treatment was still 1.5–3 times higher than the N fertilization rates in smallholder oil palm plantations, and these reduce fertilization treatment displayed 2–4 times larger soil $N_2O$ emissions than the smallholder plantations (Hassler et al., 2017). Also, the soil mineral N levels in this large-scale oil palm plantation were larger in any of the management treatments (Table 2) compared to smallholder oil palm plantations (Hassler et al., 2017). In the reduced fertilization treatment, soil mineral N was possibly not the limiting factor for $N_2O$ production, since the peaks of soil $N_2O$ emissions following fertilization in both reduced and conventional fertilization treatments were comparable (Fig. S2). This supports the conclusion of Formaglio et al. (2020) that the decadal over-fertilization of this large-scale oil palm plantation causes large stocks of mineral N, leached below the root zone, and despite four years of reduced fertilization, mineral N stored at deeper depths can contribute to microbial production of $N_2O$. These findings imply the need to adjust fertilization rates with age of oil palm plantation to maintain good yield while reducing the environmental impact. Apparently, years of over fertilization can have lasting effects well beyond the period when fertilization management changes. As the palm circle is a hotspot of $N_2O$ emissions, improved nutrient management in this zone may have the potential to minimize fertilizer-induced $N_2O$ emissions, e.g. through application of slow-release N fertilizers, use of nitrification inhibitors, adjusting N application rate with age of the plantation, and understory vegetation to take up and recycle excess mineral N (Sakata et al., 2015; Ashton-Butt et al., 2018; Cassman et al., 2019). Moreover, return of organic residues (empty fruit bunches or mill effluent) should be encouraged to improve nutrient retention and recycling, and to reduce dependency on chemical fertilizers (Bakar et al., 2011; Formaglio et al., 2021).



### 4.3 Soil CH$_4$ uptake

Area-weighted soil CH$_4$ uptake (Table 1) was comparable to CH$_4$ uptake reported for oil palm plantations ($-15 \pm 3$ µg C m$^{-2}$
h$^{-1}$) but lower than soil CH$_4$ uptake in forests on similar loam Acrisol soils in Jambi, Indonesia ($-29 \pm 12$ µg C m$^{-2}$ h$^{-1}$) (Hassler
et al., 2015). The soil CH$_4$ uptake at our site was larger than that reported by Drewer et al., (2021a) ($-3 \pm 1$ µg C m$^{-2}$ h$^{-1}$) from
$\leq 12$ years old oil palm plantations on clay Acrisol soil in Malaysian Borneo, which they attributed to very high CH$_4$ emissions
from a plot adjacent to a riparian area. Also, the soil CH$_4$ uptake at our site was larger than that reported by Aini et al. (2020)
(ranging from $-1$ to 13 µg C m$^{-2}$ h$^{-1}$) for an oil palm plantation on sandy clay loam Cambisol soil in Jambi, Indonesia, which
they explained by the high WFPS (> 80%) during their measurement period. Clay content or soil texture is the main site factor
that correlate positively to soil CH$_4$ fluxes (Veldkamp et al., 2013), indicating that the higher the clay content the lower is the
soil CH$_4$ uptake. High clay content soils have a low proportion of coarse pores (Hillel, 2003), which are important for gas
diffusive transport. Furthermore, soils with high clay content have high water-holding capacity, which hinders gas diffusion
from the atmosphere to the soil and limits CH$_4$ availability to methanotrophic activity in the soil (Keller and Reiners, 1994;
Veldkamp et al., 2013). Thus, the disparity of soil CH$_4$ uptake between our present site and these above-mentioned studies was
attributed to their differences in soil texture or drainage status as well as rainfall conditions or WFPS during the measurement
periods, which all influence CH$_4$ diffusion from the atmosphere to the soil and thereby its uptake.

The positive correlation between soil CH$_4$ fluxes and WFPS (Fig. 4g), which reflected the same spatial pattern and positive
correlation between soil CH$_4$ fluxes and soil bulk densities (Fig. S5), was attributed to the reduction of CH$_4$ diffusion from the
atmosphere to the soil with increases in WFPS and soil bulk density (Veldkamp et al., 2013; Martinson et al., 2021). Differences
in soil bulk density that result from management practices in oil palm plantations (Table S1) (Formaglio et al., 2021) affect soil
total porosity, WFPS and gas diffusivity (Keller and Reiners, 1994; Hassler et al., 2015). The frond-stacked area has large SOC
(Table S1) and low soil bulk density (Fig. S5) or high porosity (Formaglio et al., 2021), resulting in low WFPS (Table 2). Thus,
with the high soil porosity in the frond-stacked area, gas diffusion may not limit CH$_4$ availability to methanothrophic activity
in the soil, resulting in the highest soil CH$_4$ uptake among the management zones (Table 1). Where gas diffusion was favorable
in the frond-stacked area, the increase in soil mineral N had increased CH$_4$ uptake (Fig. 4i), suggesting that mineral N
availability enhanced CH$_4$ uptake once gas diffusion is not limiting (Veldkamp et al., 2013; Hassler et al., 2015). Conversely,
the palm circle and inter-row have small SOC (Table S1) and high soil bulk density (Fig. S5) or low porosity (Formaglio et al.,
2021), resulting in high WFPS (Table 2), which may have limited gas diffusion and possibly created anaerobic microsites, and
thereby the occasional soil CH$_4$ emissions during the wet season (Fig. S3). Thus, the observation that soil mineral N did not
influence soil CH$_4$ fluxes from the palm circle and inter-row (Fig. 4i) was possibly because gas diffusion limitation was the
overriding factor controlling soil CH$_4$ uptake. Aside from improving the soil biochemical properties with the decadal piling of
senesced fronds on the frond-stacked area (Table S1), which favor for increases in soil microbial biomass (Fig. S4) and N
cycling rate (Formaglio et al., 2021), stimulating soil CH$_4$ sink or methanotrophic activity is one proof of the multiple benefits
of conserving soil organic matter, e.g. its role on soil GHG abatement (Veldkamp et al., 2020). Foot traffic from management
practices in the palm circle and inter-row as well as reduced litter inputs had increased soil bulk density and decreased SOC
(Table S1), accompanied by reductions in microbial biomass and its activity (e.g. low soil N cycling rate; Formaglio et al.,
2021), including reduced methanotrophic activity (Table 1). At the plantation level, the overall comparable soil CH$_4$ uptake
between the reduced and conventional management treatments indicated that changes in fertilization rates and weeding methods





did not yet affect the drivers of soil $CH_4$ uptake (e.g. comparable soil mineral N and WFPS among treatments; Table 2) at least during the first four years of this experiment. Instead, the spatial differences in soil $CH_4$ uptake suggest that restoring the function of soil as $CH_4$ sink should be geared towards increasing soil organic matter, e.g. alternating locations of piled fronds with unused inter-rows, returning empty fruit bunches and other processing by-products, and avoiding plant biomass burning in establishing the next generation oil palm plantation (Bakar et al., 2011; Carron et al., 2015; Bessou et al., 2017).

**4.4 Global warming potential**

The GWP of this ≥ 18-year old, large-scale oil palm plantation (Fig. 3) was in the lower end of the estimate from another part of this plantation near a peat soil (GWP of 686 ± 353 g $CO_2$-eq $m^{-2}$ $yr^{-1}$; Meijide et al., 2020). The slight difference between our estimated GWP and this previous study, aside from the latter's proximity to a peat soil, can be due to plantation age, different climatic conditions during separate study years, and different employed methods. First, as to plantation age, oil palm

acts as a net C source one year after forest conversion with a net ecosystem exchange (NEE) of 1012 ± 51 g C $m^{-2}$ $yr^{-1}$ and becomes a net C sink at 12 years old (NEE of −754 ± 38 g C $m^{-2}$ $yr^{-1}$; Meijide et al., 2020). However, C removed from the field via fruit harvest turns this plantation into a net C source (NEP of 146 ± 94 g C $m^{-2}$ $yr^{-1}$; Meijide et al., 2020). At our study plots, the average annual yield during 2017–2020 across treatments (Fig. 3; Table S2) was higher than that of Meijide et al. (2020) (900 ± 49 g C $m^{-2}$ $yr^{-1}$). Our estimated NEP (i.e. heterotrophic respiration – (NPP – fruit biomass C); Fig. 3) (Quiñones

et al., 2022; Iddris et al, 2023) was in the lower end of Meijide et al.'s (2020) NEP estimate. We attributed this small NEP to be due to large biomass and yield production as oil palm trees aged and also possibly due to reduced heterotrophic respiration as SOC had already decreased and reported to attain a steady-state low level after ~15 years from forest conversion (van Straaten et al., 2015). Secondly, the different climate conditions during our study year compared to the study by Meijide et al. (2020) may also have contributed to the differences in biomass and yield production as well as heterotrophic respiration. Our

measurement period (2019–2020) had annual rainfall within the 10-year average, whereas Meijide et al.'s (2020) study years (2014–2016) included a severe drought in 2015 caused by a strong El Niño Southern Oscillation that induced prolonged smog events in Jambi, Indonesia (Field et al., 2016). Drought combined with smoke haze reduce the productivity and $CO_2$ uptake in this oil palm plantation (Stiegler et al., 2019), supporting the low fruit yield measured by Meijide et al. (2020). Thirdly, our different estimation methods may have contributed to the small disparity between our GWP estimate and that by Meijide et al.

(2020), who measured NEE by eddy covariance technique. Our method and theirs both have advantages and disadvantages; ours based on measurements of NPP components and assumption of heterotrophic respiration contribution to measured soil $CO_2$ emissions (van Straaten et al., 2011) was spatially replicated, inexpensive and practicable to deploy without a need for electricity source in the field (Baldocchi, 2014), e.g. for plot-scale experiment on different fertilization regimes and weeding practices.

As soil $N_2O$ emissions contributed substantially (55%) to the GWP of this large-scale oil palm plantation while soil $CH_4$ sink had only minor offset (< 2%) (Fig. 3), reducing its GHG footprint could be achieved by decreasing soil $N_2O$ emissions and increasing soil $CH_4$ uptake (see above). Finally, the large NPP of this ≥ 18-year old, large-scale oil palm plantation and reduced heterotrophic respiration (due to reduced SOC after ~ 15 years from forest conversion; van Straaten et al., 2015) contributed to its small NEP that accounted 46% of the GWP. In the perspective of long-term oil palm management, extending the rotation

period from 25 years to 30 years to prolong accumulation of plant biomass C (Meijide et al., 2020), avoiding large biomass



loss during establishment of the next generation oil palms (e.g. not burning but leaving cut palm trees on the field), and enhancing SOC stocks will reduce the GHG footprint of oil palm plantations.

## 5 Conclusions

During the first four years of this management experiment, soil GHG fluxes, GWP, and yield in reduced fertilization with
mechanical weeding remained similar to conventional fertilization with herbicide application, signifying the strong legacy effect of over a decade of high fertilization regime prior to the start of our experiment in this mature oil palm plantation. Reducing soil $N_2O$ emissions is the key to reducing GHG footprint in this oil palm plantation as soil $N_2O$ emissions contributed substantially to the GWP of this oil palm plantation. The palm circle and frond-stacked area both showed high mineral N availability, but the palm circle was driven by the high fertilization rates and rendering it as a hotspot of soil $N_2O$ emissions. In
contrast, the frond-stacked area support high root and microbial biomass, low soil $N_2O$ emissions and high soil $CH_4$ uptake. Thus, reasonably expanding the frond-stacked area and returning organic residues from oil palm fruit processing in order to reduce dependency on chemical fertilizers will help reduce the GHG footprint while maintaining high production in oil palm plantations.

### Acknowledgements

This study was funded by the Deutsche Forschungsgemeinschaft (DFG) project number 192626868—SFB 990 /2-3) in the framework of the Collaborative Research Center 990 EFForTS as part of project A05. Guantao Chen was supported by China Scholarship Council. The PT Perkebunan Nusantara VI company provided no funding and did not have any influence on the study design, data collection, analysis, or interpretation. We thank PTPN VI for allowing us to conduct research in their plantation. We are especially thankful to our Indonesian field and laboratory assistants, Fajar Sidik, Mohammed Fatoni and the
project Z01 field personnel, for managing the field implementation of this experiment. We thank Andrea Bauer, Dirk Böttger, Kerstin Langs, and Natalia Schröder for their assistance in the laboratory analysis. This study was conducted under the research permit 148/E5/E5.4/SIP/2019.

### Author contributions

EV and MDC conceptualized the study, implemented the field-plot design and measurement methodologies. AT, BI and MD
facilitated field access, logistical support, collaborator agreements and material exports. GC conducted the field works and data analysis, and wrote the first draft of the manuscript. MDC and EV revised extensively the manuscript, and all co-authors approved the final manuscript.

### Funding

This study was funded by the Deutsche Forschungsgemeinschaft (DFG) project ID 192626868 (SFB 990/2-3) as part of project
A05.



**Data availability**

All data of this study are deposited on the GRO Göttingen Research Online data repository (https://data.goettingen-research-online.de) and under the following DOIs: 10.25625/ NR8SRD, which is accessible to all members of the Collaborative Research Center (CRC) 990. Based on the data sharing agreement within the CRC 990, these data are currently not publicly accessible
but will be made available through a written request to the senior author.

**Declaration**

**Conflict of interest**    At least one of the (co-)authors is a member of the editorial board of Biogeosciences.

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





**Table 1. Soil CO₂, N₂O, and CH₄ fluxes (means ± SE, $n$ = 4 plots) from different fertilization and weeding treatments in an ≥ 18-year old, large-scale oil palm plantation, Jambi, Indonesia, measured monthly from July 2019 to June 2020**

| Soil greenhouse gas | Management zones | Treatments | | | | across treatments | LME model $P$-values numDF = 1, denDF = 12 | | |
|---|---|---|---|---|---|---|---|---|---|
| | | ch | cw | rh | rw | | Fertilization | Weeding | Interaction |
| CO₂ flux (mg C m⁻² h⁻¹) | Palm circle | 82.06 ± 9.95 | 84.27 ± 8.70 | 92.32 ± 12.69 | 93.51 ± 5.70 | 88.04 ± 4.48 b | 0.95 | 0.94 | 0.89 |
| | Inter-row | 38.54 ± 4.73 | 41.46 ± 3.41 | 50.34 ± 4.14 | 39.86 ± 5.31 | 42.55 ± 2.32 c | 0.31 | 0.47 | 0.13 |
| | Frond-stacked area | 126.21 ± 6.23 | 132.95 ± 11.17 | 120.23 ± 11.15 | 117.95 ± 6.57 | 124.33 ± 4.34 a | 0.21 | 0.89 | 0.65 |
| | area-weighted | 59.52 ± 3.49 | 62.60 ± 2.05 | 68.38 ± 6.37 | 61.23 ± 4.45 | 62.93 ± 2.14 | 0.55 | 0.66 | 0.22 |
| N₂O flux (µg N m⁻² h⁻¹) | Palm circle | 198.96 ± 53.70 | 219.50 ± 131.74 | 301.90 ± 121.53 (205.17 ± 85.71) [a] | 91.47 ± 7.55 [a] | 202.96 ± 46.14a | 0.38 | 0.16 | 0.31 |
| | Inter-row | 10.14 ± 3.94 | 7.40 ± 1.56 | 13.67 ± 5.56 | 4.48 ± 1.13 | 8.92 ± 1.81 c | 0.90 | 0.14 | 0.26 |
| | Frond-stacked area | 12.82 ± 0.98 | 16.12 ± 4.26 | 37.50 ± 26.14 (17.73 ± 6.98) [a] | 9.74 ± 2.91 | 19.04 ± 6.59 b | 0.38 | 0.47 | 0.62 |
| | area-weighted | 44.53 ± 8.18 | 46.56 ± 24.31 | 69.13 ± 28.67 | 20.93 ± 1.87 | 45.29 ± 9.67 | 0.53 | 0.20 | 0.07 |
| CH₄ flux (µg C m⁻² h⁻¹) | Palm circle | −2.89 ± 1.19 | −3.19 ± 1.63 | −3.74 ± 2.46 | −5.97 ± 1.70 | −3.95 ± 0.87 a | 0.26 | 0.66 | 0.29 |
| | Inter-row | −16.94 ± 5.39 | −14.06 ± 1.69 | −14.13 ± 6.27 | −14.46 ± 2.98 | −14.90 ± 2.03 b | 0.89 | 0.66 | 0.84 |
| | Frond-stacked area | −42.53 ± 1.93 | −38.89 ± 3.36 | −36.09 ± 6.59 | −44.66 ± 6.44 | −40.54 ± 2.39 c | 0.88 | 0.73 | 0.28 |
| | area-weighted | −18.25 ± 3.74 | −15.89 ± 1.42 | −15.55 ± 4.75 | −17.46 ± 2.97 | −16.79 ± 1.57 | 0.98 | 0.73 | 0.58 |

For each soil greenhouse gas, different letters within each column indicate significant differences among management zones across treatments ($2^2$ factorial ANOVA with linear mixed-effects models and Tukey HSD test at p ≤ 0.05); numDF and denDF are numerator and denominator degrees of freedom, respectively. ch: conventional fertilization – herbicide weeding, cw: conventional fertilization – mechanical weeding, rh: reduced fertilization – herbicide weeding, rw: reduced fertilization – mechanical weeding

[a] For soil N₂O fluxes, values in parenthesis excluded two extreme outliers in the palm circle (5311 and 4325 µg N m⁻² h⁻¹) and one extreme outlier in the frond-stacked area (1934 µg N m⁻² h⁻¹)



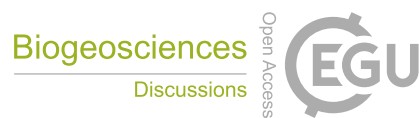

**Table 2. Soil temperature, water content, and mineral N concentrations (means ± SE, $n$ = 4 plots) measured in the top 5 cm from different fertilization and weeding treatments in an ≥ 18-year old, large-scale oil palm plantation, Jambi, Indonesia, measured monthly from July 2019 to June 2020**

| Soil factors | Management zones | Treatments | | | | across treatments | LME model $P$-values numDF = 1, denDF = 12 | | |
|---|---|---|---|---|---|---|---|---|---|
| | | ch | cw | rh | rw | | Fertilization | Weeding | Interaction |
| soil temperature (°C) | Palm circle | 26.2 ± 0.2 | 26.2 ± 0.2 | 26.3 ± 0.2 | 26.2 ± 0.1 | 26.2 ± 0.1 a | 0.44 | 0.57 | 0.55 |
| | Inter-row | 26.1 ± 0.2 | 26.2 ± 0.2 | 26.2 ± 0.1 | 26.0 ± 0.1 | 26.1 ± 0.1 a | 0.89 | 0.91 | 0.21 |
| | Frond-stacked area | 26.1 ± 0.1 | 26.2 ± 0.1 | 26.2 ± 0.1 | 26.1 ± 0.1 | 26.2 ± 0.1 a | 0.66 | 0.93 | 0.21 |
| Water-filled pore space (%) | Palm circle | 38.8 ± 2.5 | 35.6 ± 2.3 | 34.9 ± 2.9 | 40.4 ± 2.1 | 37.5 ± 1.2 a | 0.89 | 0.71 | 0.32 |
| | Inter-row | 34.4 ± 3.0 | 35.4 ± 1.4 | 36.5 ± 1.5 | 36.9 ± 1.9 | 35.8 ± 1.0 a | 0.53 | 0.71 | 0.74 |
| | Frond-stacked area | 25.2 ± 2.4 | 26.7 ± 3.0 | 27.9 ± 3.0 | 23.6 ± 1.1 | 25.8 ± 1.2 b | 0.86 | 0.53 | 0.24 |
| $NH_4^+$ ($\mu$g N g$^{-1}$) | Palm circle | 6.3 ± 5.1 | 15.2 ± 10.9 | 1.7 ± 0.8 | 13.7 ± 8.3 | 9.3 ± 3.6 a | 0.42 | 0.08 | 0.64 |
| | Inter-row | 0.7 ± 0.1 | 0.8 ± 0.1 | 0.8 ± 0.1 | 0.8 ± 0.1 | 0.7 ± 0.1 c | 0.24 | 0.84 | 0.26 |
| | Frond-stacked area | 1.8 ± 0.1 | 4.1 ± 1.9 | 2.3 ± 0.1 | 1.8 ± 0.1 | 2.50 ± 0.5 b | 0.52 | 0.27 | 0.02 |
| $NO_3^-$ ($\mu$g N g$^{-1}$) | Palm circle | 5.2 ± 2.3 | 14.6 ± 6.1 | 4.5 ± 1.9 | 10.0 ± 4.7 | 8.6 ± 2.1 a | 0.42 | 0.29 | 0.88 |
| | Inter-row | 0.4 ± 0.1 | 0.6 ± 0.2 | 0.6 ± 0.3 | 0.3 ± 0.1 | 0.5 ± 0.1 c | 0.54 | 0.39 | 0.14 |
| | Frond-stacked area | 10.1 ± 2.5 | 14.1 ± 1.4 | 5.3 ± 1.0 | 3.6 ± 0.4 | 8.3 ± 1.3 b | <0.01 | 0.53 | 0.16 |

For each soil factor, different letters within each column indicate significant differences among management zones across treatments ($2^2$ factorial ANOVA with linear mixed-effects models and Tukey HSD test at p ≤ 0.05); numDF and denDF are numerator and denominator degrees of freedom, respectively. ch: conventional fertilization – herbicide weeding, cw: conventional fertilization – mechanical weeding, rh: reduced fertilization – herbicide weeding, rw: reduced fertilization – mechanical weeding



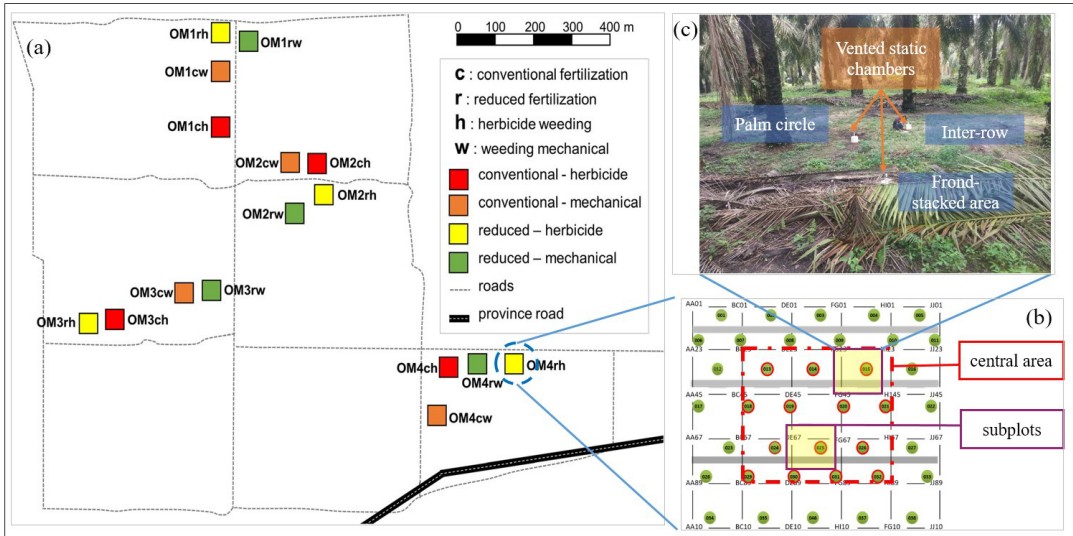

**Figure 1: Experimental set-up.** A 2 × 2 factorial experiment design with four blocks (OM1–4) within which are the four treatments (each plot was 50 m × 50 m; ch: conventional fertilization – herbicide weeding, cw: conventional fertilization – mechanical weeding, rh: reduced fertilization – herbicide weeding, rw: reduced fertilization – mechanical weeding) (a). Two subplots were selected in the central 30 m × 30 m area in each plot (b). In each subplot, soil GHG flux measurements were conducted at each management zone (palm circle, inter-row, and frond-stacked area) using vented static chambers (c)

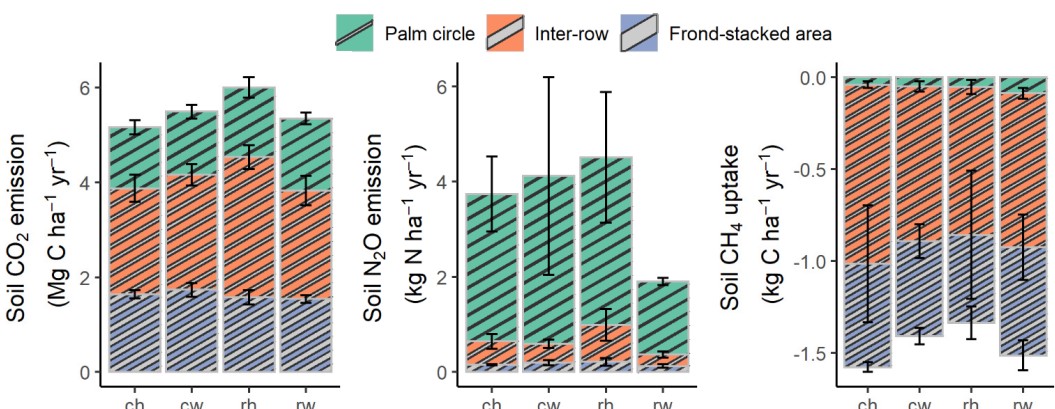

**Figure 2: Annual soil CO₂, N₂O, and CH₄ fluxes (means ± SE, *n* = 4 plots), weighted by the areal coverages of the palm circle (18%), inter-row (67%), and frond-stacked area (15%) under different fertilization and weeding treatments in an ≥ 18-year old, large-scale oil palm plantation, Jambi, Indonesia.** ch: conventional fertilization – herbicide weeding, cw: conventional fertilization – mechanical weeding, rh: reduced fertilization – herbicide weeding, rw: reduced fertilization – mechanical weeding. Annual soil N₂O emissions from rh were calculated excluding three extreme outliers: 5311 and 4325 µg N m⁻² h⁻¹ in the palm circle and 1934 µg N m⁻² h⁻¹ in the frond-stacked area. Annual soil CO₂, N₂O, and CH₄ fluxes were not statistically tested since these are trapezoidal extrapolations between measurement periods





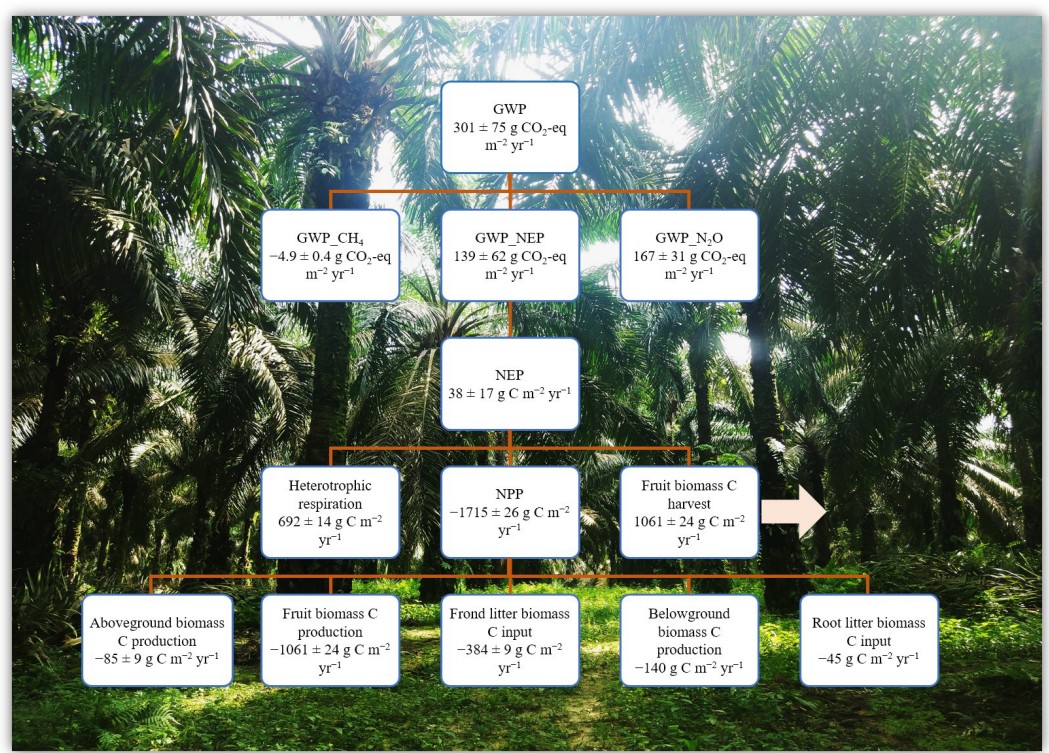

**Figure 3: Ecosystem net global warming potential (GWP) from an ≥ 18-year old, large-scale oil palm plantation,**
**Jambi, Indonesia (means ± SE, *n* = 16 plots).** Negative and positive symbols indicate the direction of the flux: (-)
for C uptake and (+) for C export or emission from the plantation. Net primary productivity (NPP, g C m$^{-2}$ yr$^{-1}$) =
aboveground biomass C production + fruit biomass C + frond litter biomass C + belowground biomass C production
+ root litter biomass C. Aboveground biomass C production = annual biomass production per palm (2019–2020) ×
planting density × tissue C concentration. Annual fruit biomass C = annual fruit production per palm (mean of 2019–
2020) × planting density × tissue C concentration. Frond litter biomass C = annual litter production per palm (mean
of 2019–2020) × planting density × tissue C concentration.  Belowground biomass C production and root litter C were
taken from oil palm sites in the same area (Kotowska et al., 2015). Net ecosystem productivity (NEP, g C m$^{-2}$ yr$^{-1}$) =
heterotrophic respiration – (NPP – fruit biomass C) (Malhi et al., 1999; Quiñones et al., 2022). Heterotrophic
respiration was assumed to be 70% of soil respiration (van Straaten et al., 2011) + 80% from decomposition of frond
litter (Iddris et al., 2023). GWP (g CO$_2$-eq m$^{-2}$ yr$^{-1}$) = NEP_in CO$_2$ + N$_2$O_CO$_2$-eq + CH$_4$_CO$_2$-eq, whereby the CO$_2$-
equivalents for N$_2$O and CH$_4$ are their annual fluxes multiplied by 298 and 25, respectively, for a 100-year time frame
(IPCC 2006)



**Figure 4: Relationships among soil CO₂, N₂O, and CH₄ fluxes and soil factors.** Spearman rank coefficients (rho)

690    marked by * indicates $p \leq 0.05$. Each data point was the average of four replicate plots per treatment on each measurement period ($n = 48$ (i.e. 4 treatments × 12 months) for soil temperature; $n = 40$ (i.e. 4 treatments × 10 months) for soil water-filled pore space (WFPS) and total mineral N). P – palm circle, IR – inter-row, FS – frond-stacked area