# Peer review of "Large contribution of soil N2O emission to the global warming potential of a large-scale oil palm plantation despite changing from conventional to reduced management practices"

_Biogeosciences, 2023_

## Referee Comment (RC1)

Review BGD-2023-102

The authors present a nicely written study on the effect of reduced fertilisation rates in combination with mechanical weeding on GHG fluxes in an industrial oil palm plantation in Indonesia. I support the publication subject to revision detailed below IF all points can be addressed. Hopefully it is just a case of clarification and not serious flaws in the study design.

My main points are that you need to clarify early on that your presented CO2 fluxes are from chambers and likely not soil respiration as the enclosure time was too long for that. Please clarify whether there was vegetation present in the chambers or soil only. This needs to be caveated throughout, especially when you compare your results to studies that measured 'proper' soil respiration.

Please clarify your measurement regime around the fertilisation period. It is not clear whether measurements were more frequent after fertilisation (which they should have). Otherwise your interpretation of fluxes due to fertilisation might be flawed and cannot be accepted for publication in its current form. To characterise peak N2O emissions after fertilisation, daily measurements are needed initially and frequent measurements at least over two weeks until fluxes are back to background levels. Otherwise no sounds cumulative emissions from a fertilisation event can be determined.

Specific comments:

L22 add 'for' after accounted

L149/150 & L173 change to the 'University of Goettingen'

L138 Did your chambers include vegetation or just soil? This particularly important to interpret the soil CO2 fluxes. You can't call them 'soil respiration' later on (e.g. l 299) if some of the chambers contained vegetation or litter such as palm fronts. A better term to use might be soil efflux or ecosystem respiration?

L140 How frequently did you measure after fertilisation? Did you measure more frequently after the fertilisation? It is not clear at the moment as figure S2 only shows monthly measurements. There is a risk you are over-interpreting your results if you only measured once after application.

Section 2.4 (L176 to L197) Please write out the equations with an equation editor, number them and then refer to them in the text. It would make it a lot clearer to see which equations have been used and what the parameters within one equation are.

Figure S2 It is not clear whether you measured more frequently after fertiliser application. Only using monthly measurements you might not have captured the peak emissions after fertilisation adequately and you cannot base statements on one measurement after fertilisation.

L 265-275 Be careful what you compare your CO2 fluxes with. Some of your referenced studies reported pure soil respiration measured from soil only with infrared gas analyser and proper soil respiration protocols. You are presenting chamber measurements using a different technique and potentially vegetation present in your chambers. So please add a caveat to this part of your discussion.

L331 If you have only measured once after fertilisation, your entire argument might be flawed.

L 337 change to 'reduced'

L429 You only measured GHG for one year so concluding here over 4 years is a bit misleading, if you are including results from other studies in this statement please mention it

---

## Author Comment (AC1)

**Answers to feedbacks from Reviewer 1**

We would like to thank sincerely reviewer 1 for her/his comments that help improve our manuscript (BGD-2023-102). We have indicated below our replies to the comments of reviewer 1. When we mentioned the line numbers, where revisions are made, these line numbers are in the revised manuscript without the tracked changes.

1.1. The authors present a nicely written study on the effect of reduced fertilization rates in combination with mechanical weeding on GHG fluxes in an industrial oil palm plantation in Indonesia. I support the publication subject to revision detailed below IF all points can be addressed. Hopefully it is just a case of clarification and not serious flaws in the study design.

Thanks for your time and helpful comments that improve our manuscript.

1.2. My main points are that you need to clarify early on that your presented $CO_2$ fluxes are from chambers and likely not soil respiration as the enclosure time was too long for that. Please clarify whether there was vegetation present in the chambers or soil only. This needs to be caveated throughout, especially when you compare your results to studies that measured 'proper' soil respiration.

The method of soil GHG measurement was clearly stated in section 2.2 first sentence. Also, please see our answers to reviewer 1's questions #1.6 and #1.10 below.

We added the sentence "Any aboveground vegetation inside the chambers was carefully cut during the study period but root and litter remained as normal" (lines 148-149 of the revised manuscript).

1.3. Please clarify your measurement regime around the fertilisation period. It is not clear whether measurements were more frequent after fertilisation (which they should have). Otherwise your interpretation of fluxes due to fertilisation might be flawed and cannot be accepted for publication in its current form. To characterise peak N2O emissions after fertilisation, daily measurements are needed initially and frequent measurements at least over two weeks until fluxes are back to background levels. Otherwise no sounds cumulative emissions from a fertilisation event can be determined.

We have characterized in details the peaks of soil $N_2O$ emissions following fertilization in our earlier studies in both smallholder and large-scale oil palm plantations (Hassler et al., 2017; Meijide et al., 2020). From our previous studies, we knew that the peaks of soil $N_2O$ emissions occurred within 2 weeks following fertilization. Thus, as stated in the second sentence of section 2.2., we conducted the measurements in the palm circles (with 2 chambers in each of the 16 replicate plots) within 2 weeks following the fertilization (lines 140-141 of the original manuscript). Fertilization was conducted only twice a year (in April and October of each year; line 116 of the original manuscript), and because fertilization was only twice a year, sampling schedule that employed frequent measurements following fertilization did not show significant differences in soil $N_2O$ emissions as compared to one sampling schedule that captured the peak of $N_2O$ emission within 2 weeks following fertilization (please see Hassler et al., 2017 – first paragraph under section 4.3).

We agree with the reviewer 1's comment that capturing the peak of $N_2O$ emission is important, and we have captured these peak emissions with our one measurement period within 2 weeks following fertilization, as stated in the original manuscript: line 223, Fig.

S2, line 232, Table 1, lines 235-236, Fig. 2. We argue that it is important that our sampling design not only captured the $N_2O$ peak from fertilization (Fig. S2) but more importantly also represented the spatial variation brought by management practices, representing the fertilized palm circle (only 18% of the area), unfertilized inter-row (67% of the area) and frond-stacked area (15% of the area) (Fig. 2). As we have discussed in the last paragraph of section 4.2, the short-term fertilization effect may not be the most important in determining the annual $N_2O$ emission in this mature oil-palm plantation but instead the long-term legacy effects (more than a decade) of conventional high fertilization, prior to the start of this management experiment. The 3-4 years of reduced fertilization did not yet affect the soil $N_2O$ fluxes, as also supported by the comparable mineral N levels between conventional and reduced fertilization (Table 2).

Moreover, we would like to point out that our measurement period (July 2019-June 2020) fell during the prolonged covid-19 lockdown (March 2020-May 2022) in Indonesia. There were huge logistical difficulties in continuing our original plan of conducting intensive measurement following fertilization, for validation of our previous findings. Despite the lockdown, we managed to continue the measurement regime to capture the $N_2O$ peak within 2 weeks following fertilization on the palm circle (Fig. S2) and all the rest of the study year with monthly measurement as permitted during the lockdown (i.e. reduced number of physical presence of personnel) – 16 plots × 2 subplots/plot × 3 management zones/subplot (palm circle, inter-row, frond-stacked area).

Our data are valuable as there had been no full accounting of soil GHG fluxes from spatially replicated management experiment in a large-scale oil palm plantation.

1.4. L22 add 'for' after accounted

We added "for" after accounted.

1.5. L149/150 & L173 change to the 'University of Goettingen'

Thank you for pointing this out. We changed "Goettingen University" to "University of Goettingen"

1.6. L138 Did your chambers include vegetation or just soil? This particularly important to interpret the soil CO2 fluxes. You can't call them 'soil respiration' later on (e.g. l 299) if some of the chambers contained vegetation or litter such as palm fronds. A better term to use might be soil efflux or ecosystem respiration?

Our chambers had no aboveground vegetation inside but the roots and litter were left as in normal condition. It is wrong to use ecosystem respiration because ecosystem is both heterotrophic and autotrophic (roots and aboveground plant respiration; Malhi et al. 1999). We think soil respiration is appropriately used for chamber-based measurements that include both root and heterotrophic respiration (as roots remain at depths in the soil even if the aboveground vegetative parts inside the chamber are carefully cut-off). Please also see our answers to reviewer 1's question #1.10 below (Trumbore, 2006; see Figure below).

Nonetheless, in order to accommodate the reviewer 1's concern, we now used 'soil $CO_2$ efflux' to replace soil respiration all throughout the revised manuscript.

1.7. L140 How frequently did you measure after fertilisation? Did you measure more frequently after the fertilisation? It is not clear at the moment as figure S2 only shows monthly measurements. There is a risk you are over-interpreting your results if you only measured once after application.

Please see our answers to question #1.3 above.

1.8. Section 2.4 (L176 to L197) Please write out the equations with an equation editor, number them and then refer to them in the text. It would make it a lot clearer to see which equations have been used and what the parameters within one equation are.

Thanks for your suggestion. But before reading below, please see also section 2. 1 second paragraph to refresh the experimental plot design. We addressed this suggestion of reviewer 1 by rewriting this part with equations.

First, the net primary production (NPP) was determined. Within the inner 30 m × 30 m area in each replicate plot (Fig. 1b), all the palms were measured for their stem height, harvested fruit weight, and the number of pruned fronds during 2017–2020 (Iddris et al., 2023). Aboveground biomass per palm was calculated using the allometric growth equation of Asari et al. (2013). Annual aboveground biomass production per palm is the difference in aboveground biomass between two consecutive years, averaged over a two-year period (2018-2019 and 2019-2020).

Aboveground biomass C production (g C $m^{-2}$ $yr^{-1}$) = annual aboveground biomass production per palm (kg $palm^{-1}$) × planting density (142 palms $ha^{-1}$) × tissue C concentration (0.41 g C $g^{-1}$) × $10^{-1}$ (for unit conversion)    (1)

Fruit biomass C production (g C $m^{-2}$ $yr^{-1}$) = annual fruit harvest per palm (kg $palm^{-1}$, mean of 2019–2020) × planting density × tissue C concentration (0.63 g C $g^{-1}$) × $10^{-1}$ (for unit conversion)    (2)

Frond litter biomass C input (g C $m^{-2}$ $yr^{-1}$) = annual litter production per palm (kg $palm^{-1}$, mean of 2019–2020) × planting density × tissue C concentration (0.47 g C $g^{-1}$) × $10^{-1}$ (for unit conversion)    (3)

NPP (g C $m^{-2}$ $yr^{-1}$) = Aboveground biomass C production + Fruit biomass C production + Frond litter biomass C input + Root biomass C production (140 g C $m^{-2}$ $yr^{-1}$; Kotowska et al. 2015) + Root litter biomass C input (45 g C $m^{-2}$ $yr^{-1}$; Kotowska et al. 2015)    (4)

Second, the net ecosystem productivity (NEP) was calculated following Malhi et al. (1999) and Quiñones et al. (2022) for agricultural land use.

NEP (g C $m^{-2}$ $yr^{-1}$) = heterotrophic respiration – (NPP – fruit biomass C)    (5)

Our measured soil $CO_2$ fluxes included both autotrophic and heterotrophic respirations. We assumed 70% heterotrophic contribution to soil $CO_2$ efflux, based on a long-term quantification in a forest in Sulawesi, Indonesia (van Straaten et al., 2011). As the frond litter also contributes to heterotrophic respiration upon decomposition, we assumed this fraction to be 80% of frond litter biomass C, based on the frond-litter decomposition rate in the same plantation (Iddris et al., 2023). We used the area-weighted value (based on the areal coverages of the three management zones; see 2.1 above) of the annual heterotrophic respiration to calculate NEP for each replicate plot.

Third, the GWP was calculated following Meijide et al. (2020) and Quiñones et al. (2022).

GWP (g $CO_2$-eq $m^{-2}$ $yr^{-1}$) = (NEP × 3.67) + (soil $N_2O$ fluxes × 298) + (soil $CH_4$ fluxes × 25)    (6)

whereby 3.67 is C-to-$CO_2$ conversion, and 298 and 25 are $CO_2$-equivalents of $N_2O$ and $CH_4$, respectively, for a 100-year time horizon (IPCC, 2006). Similarly, we used the area-weighted values of the annual soil $N_2O$ and $CH_4$ fluxes (see 2.2 above) to calculate GWP for each replicate plot. Negative and positive symbols indicate the direction of the flux: (−) for C uptake

and (+) for C export or emission from the plantation.

1.9. Figure S2 It is not clear whether you measured more frequently after fertiliser application. Only using monthly measurements you might not have captured the peak emissions after fertilisation adequately and you cannot base statements on one measurement after fertilisation.

Please see the answers to #1.3 above.

1.10. L 265-275 Be careful what you compare your CO2 fluxes with. Some of your referenced studies reported pure soil respiration measured from soil only with infrared gas analyser and proper soil respiration protocols. You are presenting chamber measurements using a different technique and potentially vegetation present in your chambers. So please add a caveat to this part of your discussion.

As we mentioned in 1.6 above, we removed the aboveground vegetation inside the chamber but the roots at soil depths were not disturbed, which contribute to soil $CO_2$ efflux. Please refer to Malhi et al. (1999) and Trumbore (2006; see Figure below) the definition of soil respiration = autotrophic + heterotrophic respiration.

1.11. L331 If you have only measured once after fertilisation, your entire argument might be flawed

Please see the answers to reviewer 1's question #1.3 above.

1.12. L 337 change to 'reduced'

Thank you for pointing this out. We changed this.

1.13. L429 You only measured GHG for one year so concluding here over 4 years is a bit misleading, if you are including results from other studies in this statement please mention it.

Thank you for pointing this out. What we meant is the first four years of this oil palm management experiment. Of course the soil GHG fluxes were measured only during 2.5-3.5 years of the experiment whereas the palm yields and biomass were measured since the start of this experiment through 2020 (covered in this study) until now (beyond this manuscript's study period).

To avoid confusion, we changed the sentence to:

"During the 3-4 years of this management experiment, soil GHG fluxes, GWP, and yield in reduced fertilization with mechanical weeding remained similar to conventional fertilization with herbicide application, signifying the strong legacy effect of over a decade of high fertilization regime prior to the start of our experiment in this mature oil palm plantation."

References

Hassler, E., Corre, M. D., Kurniawan, S., and Veldkamp, E.: Soil nitrogen oxide fluxes from lowland forests converted to smallholder rubber and oil palm plantations in Sumatra, Indonesia, Biogeosciences, 14, 2781–2798, https://doi.org/10.5194/bg-14-2781-2017, 2017.

Malhi, Y., Baldocchi, D. D., and Jarvis, P. G.: The carbon balance of tropical, temperate and boreal forests, Plant Cell Environ, 22, 715–740, https://doi.org/10.1046/j.1365-3040.1999.00453.x, 1999.

Meijide, A., de la Rua, C., Guillaume, T., Röll, A., Hassler, E., Stiegler, C., Tjoa, A., June, T., Corre, M. D., Veldkamp, E., and Knohl, A.: Measured greenhouse gas budgets challenge emission savings from palm-oil biodiesel, Nat Commun, 11, 1089, https://doi.org/10.1038/s41467-020-14852-6, 2020.

Trumbore, S.: Carbon respired by terrestrial ecosystems – recent progress and challenges. Global Change Biology, 12,141–153, doi: 10.1111/j.1365-2486.2005.01067.x, 2006

---

## Author Comment (AC2)

**Answers to feedbacks from Reviewer 2**

We would like to thank sincerely reviewer 2 for her/his comments, which greatly improve our manuscript (BGD-2023-102). We have indicated below our replies to the comments of reviewer 2. When we mentioned the line numbers, where revisions are made, these line numbers are in the revised manuscript without the tracked changes.

1.1 The manuscript submitted by G. Chen and colleagues describes a very nice study on the determination of emissions of the main greenhouse gases (GHG) $CO_2$, $N_2O$ and $CH_4$ and the related Global Warming Potential (GWP) from an oil palm plantation in Indonesia. The experimental setup and statistical analyses appear to be sound and the results are presented in a nice and understandable way.

Thank you so much for your positive comments.

1.2 I think that this study has the potential to be published in Biogeosciences – there are, however, some things that should be re-worked and re-written. In general, it is not always clear to me which of the results have been obtained in the manuscript presented here and which have already been reported and published in previous studies. From what I understand have different publications come out of the experiment (which is great!) and the GHG measurements are part of this particular manuscript. Sentences such as ll. 317 – 318 make it difficult to understand if the emission peaks and pulses have been observed in this study (the material is presented in the supplementary material) or in the publications that are cited.

You are right, it seems not clear enough for readers. The soil GHG fluxes are primary data in this present study, not reported elsewhere (only the annual soil GHG values were used by Iddris et al. (2023), but these were transformed into standardized Z-values, as two of the many indicators on ecosystem functions in the overall synthesis of the oil palm management experiment).

For sentences in ll.317--318 of the original manuscript, we rewrote as follows in lines 321-324 of the revised manuscript:

"These pulses of soil $N_2O$ emissions usually peaked at around two weeks following N fertilization and went down to the background emissions after at most eight weeks (Aini et al., 2015; Hassler et al., 2017; Rahman et al., 2019). In our present study, we have captured these peaks of soil $N_2O$ emissions two weeks following fertilization on the palm circle and these elevated $N_2O$ emissions remained within two months from fertilization (Fig. S2)."

1.3 Please repeat hypotheses in the Results and Discussion sections (e.g., l. 281, l. 289, l. 316, l. 332). As the hypotheses are not numbered in the Introduction (ll. 95 – 100), it is not easy to re-call which of them was first and second when coming to the Results and Discussion.

We now numbered the hypotheses in the Introduction of the revised manuscript (lines 96, 99, 101, and 102) to make it clearer to the readers and specifically mentioned these numbers in the Discussion of the revised manuscript (lines 288, 291, 294, 324, 328, 382, 388, and 397).

We rewrote the hypotheses in the Introduction:

Thus, we tested these hypotheses: (1) during 2.5–3.5 years of this management experiment, the reduced fertilization with mechanical weeding will have comparable soil $CO_2$ and $CH_4$ fluxes but lower soil $N_2O$ emissions than the conventional fertilization with herbicide weeding. (2) The three management zones will differ in soil GHG fluxes, reflecting their inherent soil characteristics. Specifically, (2a) the fertilized palm circle that has high soil bulk density and root biomass but low SOC, microbial biomass and soil N cycling rate (Dassou et al., 2021; Formaglio et al., 2021) will have large soil $CO_2$ and $N_2O$ emissions but small soil $CH_4$ uptake; (2b) the unfertilized inter-row that has high soil bulk density but low SOC, microbial biomass and soil N cycling rate (Formaglio et al., 2021) will have small soil $CO_2$, $N_2O$ emissions and $CH_4$ uptake; (2c) the frond-stacked area (i.e. unfertilized but piled with pruned fronds) that has large SOC, microbial biomass and soil N cycling rate but low soil bulk density (Formaglio et al., 2021) will have large soil $CO_2$ emissions and $CH_4$ uptake but small soil $N_2O$ emissions.

1.4 In the Conclusions, I am missing some advice and future outlook, e.g., with regard to farmers. The suggestions presented originate mainly from another study on oil palm planation on a slightly different soil. What is the outcome of this study and how can/should this be used in practice?

The take home messages for farmers, plantation managers, extension workers and other stakeholders are given not only in the Conclusion but also in the summarizing (last) statements in the Discussion sections.

section 4.2 lines 352-360:

"These findings imply the need to adjust fertilization rates with age of oil palm plantation to maintain good yield while reducing the environmental impact. Apparently, years of over fertilization can have lasting effects on soil $N_2O$ emission well beyond the period when fertilization management changes. As the palm circle is a hotspot of $N_2O$ emissions, improved nutrient management in this zone may have the potential to minimize fertilizer-induced $N_2O$ emissions, e.g. through application of slow-release N fertilizers, use of nitrification inhibitors, adjusting N application rate with age of the plantation, and understory vegetation to take up and recycle excess mineral N (Sakata et al., 2015; Ashton-Butt et al., 2018; Cassman et al., 2019). Moreover, return of organic residues (empty fruit bunches or mill effluent) should be encouraged to improve nutrient retention and recycling, and to reduce dependency on chemical fertilizers (Bakar et al., 2011; Formaglio et al., 2021)."

section 4.3 lines 400-403:

"Instead, the spatial differences in soil $CH_4$ uptake suggest that restoring the function of soil as $CH_4$ sink should be geared towards increasing soil organic matter, e.g. alternating locations of piled fronds with unused inter-rows, returning empty fruit bunches and other processing by-products, and avoiding plant biomass burning in establishing the next generation oil palm plantation (Bakar et al., 2011; Carron et al., 2015; Bessou et al., 2017)."

section 4.4 lines 432-435:

"In the perspective of long-term oil palm management, extending the rotation period from 25 years to 30 years to prolong accumulation of plant biomass C (Meijide et al., 2020), avoiding large biomass loss during establishment of the next generation oil palms (e.g. not burning but leaving cut palm trees on the field), and enhancing SOC stocks will reduce the GHG footprint of oil palm plantations."

We now summarized these 'advices' in the Conclusion:

lines 442-445:

"Thus, improved nutrient management in this zone can minimize fertilizer-induced $N_2O$ emissions, e.g. through application of slow-release N fertilizers, use of nitrification inhibitors, adjusting N application rate with age of the plantation, and understory vegetation to take up and recycle excess mineral N."

lines 448-451:

"The GHG footprint of the next generation oil palm plantation can be reduced by extending the rotation period from beyond the common practice of 25 years to prolong accumulation of plant biomass C (Meijide et al., 2020), and by not burning palm biomass but leaving cut palm trees on the field during establishment of the succeeding oil palms to minimize biomass-C and SOC losses."

1.5 ll. 93 – 97: I cannot follow the hypothesis here: if gross N mineralization, microbial and root biomass have been observed to be the same between the two management treatments, why would the authors expect a decrease in N2O emissions? And at the same time no change in CO2 and CH4?

The basis of this hypothesis is discussed in details in Introduction paragraphs 2 and 3. The comparable gross rates of soil N cycling (indicative of the extant mineral N availability in the soil that influences methanotrophic activity) (Hassler et al., 2015), microbial biomass (as an index for heterotrophic activity - soil $CO_2$) (Formaglio et al., 2021; Hassler et al., 2015), and root biomass (a surrogate variable for autotrophic activity - soil $CO_2$) (Ryadin et al., 2022) were comparable among treatments. These indices are controlling factors of soil $CO_2$ and $CH_4$ fluxes. Thus, we hypothesized that soil $CO_2$ and $CH_4$ fluxes will be comparable among management treatments as these indices did not differ.

For soil $N_2O$ emission, the different rates of N fertilization (conventional of 260 kg N/ha/yr

vs reduced/compensatory to harvest export of 136 kg N/ha/yr) can induced a different direct emission effect from added N. That's based on our previous finding that the smallholder oil palm plantation with 2-4 times lower N fertilization rates than the large-scale plantation had a tendency to have lower soil $N_2O$ emission (Hassler et al. 2017). Thus, we hypothesized a decrease in soil $N_2O$ emission in reduced N fertilization. This comment is maybe related to reviewer 2's comments in #1.12; please also see our answer to #1.12.

As discussed in section 4.2 second paragraph, this hypothesis 1 for $N_2O$ was not supported by our findings, possibly due to the legacy effect of conventional high N fertilization rates for more than a decade prior to the start of our management experiment in 2016.

1.6 The definition of the management zones is not entirely clear to me: The palm circle was the area between the palm trees, but within the fertilized and managed area. How close to the trees were the measurements made? Was root respiration included? I assume that the fronds have been removed before measuring in the frond-stacked area? When has this happened? Only during the measurements or have they been removed constantly?

In line 115-120, first paragraph of M&M section 2.1, we clearly described the three management zones and referred to Fig 1 to show the three management zones and the locations of chamber bases. The palm circle (a 2 m radius from the palm base) is the zone where fertilizers and lime are applied (in April and October of each year) and is weeded every three months; this represents 18% of the plantation area. Our chamber base was placed at 1.7 m from the palm base, which was within the band fertilizers applied around the base of the palm. Yes, the root as well as the heterotrophic respiration were included in the measurements of soil $CO_2$ efflux.

The frond-stacked area is where the cut senesced fronds are piled and is neither fertilized nor weeded; this represents 15% of the plantation area. The chamber base on the frond-stacked area was kept covered by undecomposed senesced fronds (as these are yet loosely piled on the ground and not possible for the chamber base to include) but the undecomposed fronds were pushed gently aside only during the 28-minute chamber closure for soil GHG flux measurement.

The inter-row is basically the area that is outside the palm circle and not used for piling cut fronds. These inter-rows are unfertilized but weeded every six months, because these are mainly used by the workers for access during harvest and other works. The inter-row represents 67% of the plantation area.

These descriptions are incorporated in the revised manuscript, lines 147-150:

The chamber base in the palm circle was placed at 1.7 m away from the oil palm tree and the chamber base in the inter-row was about 5 m away from the oil palm trees (Fig. 1). Any aboveground vegetation inside the chambers was carefully cut during the study period but root and litter remained as normal. The chamber base in the frond-stacked area was kept covered with senesced fronds expect during the time of soil GHG flux measurement.

1.7 l. 143: 0.02 m appears to be a very shallow depth for GHG measurements and usually, a depth of 0.01 m (i.e., 10 cm) is recommended. How high were the bases and why was this depth chosen?

Thanks for your question. Our chamber base height is 11 cm (i.e., 0.11 m). In the field, the chamber base was inserted into the soil at about 2 cm (i.e., 0.02 m), and thus the chamber base height was about 9 cm above the soil surface. Similar depth of insertion was used in all our previous works on soil GHG flux measurement in Indonesia (e.g. Hassler et al., 2015; 2017; van Straaten et al., 2011).

We know that many studies insert the chamber base into soil at a depth of 10 cm (i.e., 0.1 cm), but we think that is not a good way. The reason why we only inserted the chamber base into soil to about 0.02 m is that we do not want to cut down too many roots that can greatly alter the soil water cycling inside the chamber base.

In the revised manuscript section 2.2, we added "That depth is sufficient to fix the chamber base while avoiding cutting off roots." in lines 145-146.

1.8 l. 249: each = any of the (?)

We changed the word to 'any of the'; thank you.

1.9 l. 276: if SOC, microbial and root biomass were explanatory for soil CO2 fluxes, this should be presented in the results already. In fact, the following lines until l. 282 should be moved to the Results section – also, as this supports one of the hypotheses

The data of soil SOC, microbial biomass and root biomass were from previously published studies but measured from the same experimental plots. Thus, we cannot repeat those data in our results section, but instead used them to tie our Discussion.

In order to make it clearer on which data were previously reported, we revised this by adding the references to which the primary data on soil $CO_2$ were related to previously measured soil parameters, SOC, microbial and root biomass.

in lines 280-283:

"The three management zones differed in soil $CO_2$ fluxes caused by their differences in SOC (Table S1; Formaglio et al., 2020), microbial biomass (Fig. S4; Formaglio et al., 2021) as drivers of heterotrophic respiration, and root biomass (Nelson et al., 2014) that influences autotrophic respiration. "

In the fourth paragraph in Introduction (lines 79-84), we have indeed mentioned those same variables to serve as the basis of our hypothesis 2.

"In the palm circle and inter-row, frequent management activities (weeding, pruning and harvesting) result in soil compaction by foot traffic (increased soil bulk density) and the low litter input in these zones exhibits low SOC and microbial biomass, and low soil N

cycling rate (Formaglio et al., 2021). Additionally, root biomass is high in the palm circle (Dassou et al., 2021). In the frond-stacked area, decomposition of fronds results in large SOC (with decreased soil bulk density), large microbial and fine root biomass, and high soil N cycling rate (Moradi et al., 2014; Rüegg et al., 2019; Formaglio et al., 2020; Dassou et al., 2021). Overall, the differences in soil properties and root biomass among these spatially distinct management zones (Formaglio et al., 2021) potentially drive the spatial variation of soil GHG fluxes from oil palm plantations (Hassler et al., 2015, 2017; Aini et al., 2020). Thus, estimating soil GHG emissions from oil palm plantations should take into account the spatial variability among management zones within a site or plot. "

1.10 l. 336: reduced

We corrected it; thank you.

1.11 l. 340: remove "possibly" – the authors provide the explanation in the following sentence

We have corrected this.

1.12 l. 342 – 344: I don´t understand why the this should be a result of decadal over-fertilization – in this study, the authors used higher amounts if fertilizer than used in smallholder oil palm plantations (this is mentioned in ll. 336 – 337). It should thus not be considered a legacy effect of past fertilization management

Prior to the start of this oil palm management experiment, i.e. before Nov. 2016, all the experimental plots were under conventional management – high fertilizations and herbicides - common to large-scale plantations. This present study was in the large-scale plantation (described in M&M 2.1).

The reduced fertilization rate (136 kg/ha/yr) were equal to the nutrients exported by fruit harvest (Formaglio et al., 2020; 2021) as opposed to the conventional fertilization rate of large-scale plantation of 260 kg N/ha/yr. Our previous studies were in smallholder plantations which had only 48-88 (Hassler et al., 2015; 2017), and had also much lower yield (Kotowska et al. 2015) than the large-scale oil palm plantation (Meijide et al. 2020).

Our study years encompassed only the 2.5-3.5 yrs since the start of this management experiment in this >18 yrs old oil palm plantation, and previous to the start of this management experiment, these plots have been under over fertilization for more than a decade. Considering all the measured parameters from which we based our argument in section 4.2 second paragraph, those parameters suggest substantial legacy effects of decadal conventional high fertilization rates.

1.13 l. 346: lasting effects on what?

We revised this sentence to:

"Apparently, years of over fertilization can have lasting effects on soil $N_2O$ emission well beyond the period when fertilization management changes."

1.14 l. 354 – 356: why this comparison? I would think that a plantation cannot be considered a forest

Here we compared the soil $CH_4$ fluxes between oil palm plantations and forests because some readers may be interested in that, as soil CH4 sink or source is one of the indicators of ecosystem function such as GHG regulation. As we mentioned in the Introduction:

"Oil palm expansion drives tropical deforestation (Vijay et al., 2016) and is accompanied by serious reductions in multiple ecosystem functions." (in lines 29-30)

1.15 l. 396: please repeat the GWP obtained in this study

We revised this sentence to:

"The GWP of this $\geq$ 18-year old, large-scale oil palm plantation (GWP of $301 \pm 75$ g $CO_2$-eq m$^{-2}$ yr$^{-1}$; Fig. 3) was in the lower end of the estimate from another part of this plantation near a peat soil (GWP of $686 \pm 353$ g $CO_2$-eq m$^{-2}$ yr$^{-1}$; Meijide et al., 2020)"

1.16 l. 429 – 431: I don´t think that this is a legacy effects – the fertilization is still rather high

Please see our answers for question 1.12.

References:

Formaglio, G., Veldkamp, E., Damris, M., Tjoa, A., and Corre, M. D.: Mulching with pruned fronds promotes the internal soil N cycling and soil fertility in a large-scale oil palm plantation, Biogeochemistry, 154, 63–80, https://doi.org/10.1007/s10533-021-00798-4, 2021.

Hassler, E., Corre, M. D., Tjoa, A., Damris, M., Utami, S. R., and Veldkamp, E.: Soil fertility controls soil-atmosphere carbon dioxide and methane fluxes in a tropical landscape converted from lowland forest to rubber and oil palm plantations, Biogeosciences, 12, 5831–5852, https://doi.org/10.5194/bg-12-5831-2015, 2015.

Hassler, E., Corre, M. D., Kurniawan, S., and Veldkamp, E.: Soil nitrogen oxide fluxes from lowland forests converted to smallholder rubber and oil palm plantations in Sumatra, Indonesia, Biogeosciences, 14, 2781–2798, https://doi.org/10.5194/bg-14-2781-2017, 2017.

Kotowska, M. M., Leuschner, C., Triadiati, T., Meriem, S., and Hertel, D.: Quantifying above- and belowground biomass carbon loss with forest conversion in tropical lowlands of Sumatra (Indonesia), Glob Chang Biol, 21, 3620–3634, https://doi.org/10.1111/gcb.12979, 2015.

Meijide, A., de la Rua, C., Guillaume, T., Röll, A., Hassler, E., Stiegler, C., Tjoa, A., June, T., Corre, M. D., Veldkamp, E., and Knohl, A.: Measured greenhouse gas budgets challenge emission savings from palm-oil biodiesel, Nat Commun, 11, 1089, https://doi.org/10.1038/s41467-020-14852-6, 2020.

Iddris, N. A., Formaglio, G., Paul, C., von Groß, V., Chen, G., Angulo-Rubiano, A., Berkelmann, D., Brambach, F., Darras, K. F. A., Krashevska, V., Potapov, A., Wenzel, A., Irawan, B., Damris, M., Daniel, R., Grass, I., Kreft, H., Scheu, S., Tscharntke, T., Tjoa, A., Veldkamp, E., and Corre, M. D.: Mechanical weeding enhances ecosystem multifunctionality and profit in industrial oil palm, Nat Sustain. https://doi.org/10.1038/s41893-023-01076-x, 2023.

Ryadin, A. R., Janz, D., Schneider, D., Tjoa, A., Irawan, B., Daniel, R., and Polle, A.: Early effects of fertilizer and herbicide reduction on root-associated biota in oil palm plantations, Agronomy, 12, 199, https://doi.org/10.3390/agronomy12010199, 2022.

van Straaten, O., Veldkamp, E., and Corre, M. D.: Simulated drought reduces soil $CO_2$ efflux and production in a tropical forest in Sulawesi, Indonesia, Ecosphere, 2, art119, https://doi.org/10.1890/ES11-00079.1, 2011.